# High Dose of Dietary Nicotinamide Riboside Induces Glucose Intolerance and White Adipose Tissue Dysfunction in Mice Fed a Mildly Obesogenic Diet

**DOI:** 10.3390/nu11102439

**Published:** 2019-10-13

**Authors:** Wenbiao Shi, Maria A. Hegeman, Atanaska Doncheva, Melissa Bekkenkamp-Grovenstein, Vincent C. J. de Boer, Jaap Keijer

**Affiliations:** Human and Animal Physiology, Wageningen University, PO Box 338, 6700 AH Wageningen, The Netherlands; ws456@cornell.edu (W.S.); m.a.hegeman@uu.nl (M.A.H.); atanaska.doncheva@outlook.com (A.D.); melissa.bekkenkamp-grovenstein@wur.nl (M.B.-G.); vincent.deboer@wur.nl (V.C.J.d.B.)

**Keywords:** vitamin B3, supplementation, NAD^+^, adipose tissue, Nnt, glucose tolerance

## Abstract

Nicotinamide riboside (NR) is a nicotinamide adenine dinucleotide (NAD^+^) precursor vitamin. The scarce reports on the adverse effects on metabolic health of supplementation with high-dose NR warrant substantiation. Here, we aimed to examine the physiological responses to high-dose NR supplementation in the context of a mildly obesogenic diet and to substantiate this with molecular data. An 18-week dietary intervention was conducted in male C57BL/6JRccHsd mice, in which a diet with 9000 mg NR per kg diet (high NR) was compared to a diet with NR at the recommended vitamin B3 level (control NR). Both diets were mildly obesogenic (40 en% fat). Metabolic flexibility and glucose tolerance were analyzed and immunoblotting, qRT-PCR and histology of epididymal white adipose tissue (eWAT) were performed. Mice fed with high NR showed a reduced metabolic flexibility, a lower glucose clearance rate and aggravated systemic insulin resistance. This was consistent with molecular and morphological changes in eWAT, including sirtuin 1 (SIRT1)-mediated PPARγ (proliferator-activated receptor γ) repression, downregulated AKT/glucose transporter type 4 (GLUT4) signaling, an increased number of crown-like structures and macrophages, and an upregulation of pro-inflammatory gene markers. In conclusion, high-dose NR induces the onset of WAT dysfunction, which may in part explain the deterioration of metabolic health.

## 1. Introduction

Nicotinamide adenine dinucleotide (NAD^+^) is an essential metabolic co-factor that supports proper cell functioning. In humans, long-term inadequate intake of the NAD^+^ precursors vitamin B3 and tryptophan (Trp) can cause NAD^+^ decline in multiple organs and ultimately lead to Pellagra, a disease of vitamin B3 deficiency [1]. Apart from vitamin B3 deficiency, NAD^+^ levels can also be influenced by other nutritional conditions. In mice, long-term high fat (HF) diet feeding reduced tissue NAD^+^ levels, leading to compromised metabolic performance, such as blunted metabolic flexibility and insulin sensitivity [2,3,4,5]. In obese individuals, reduced gene expression of the NAD^+^ synthesis pathway enzymes as well as NAD^+^-dependent enzymes was observed, which was associated with impaired metabolic health; e.g., insulin resistance and dyslipidemia [6]. The central role of NAD^+^ in the link between obesity and its associated metabolic dysfunctions is not surprising, because NAD^+^, NADH, nicotinamide adenine dinucleotide phosphate (NADP^+^) and NADPH serve as coenzymes in fuel oxidation, energy production, biosynthetic reactions and detoxification of reactive oxygen species. In addition, NAD^+^ acts as an essential substrate for several enzymatic reactions. One of the key players in these reactions are the sirtuin family of NAD^+^-dependent deacetylases, with important roles in the regulation of metabolic disease [7]. 

Multiple strategies that aimed to boost NAD^+^ levels have been shown to enhance sirtuin activity, including NAD^+^ precursor supplementation, the activation of NAD^+^ biosynthetic enzymes and inhibition of NAD^+^ consumers [7]. Supplementation with NAD^+^ precursors has been widely studied and seems to be the most promising strategy to boost NAD^+^ levels. NAD^+^ can be synthesized in vivo from several precursors via distinct pathways, including Trp and various forms of vitamin B3 [7]. These precursors have different properties with respect to their contribution to the NAD^+^ pool and their functional consequences. Trp contributes much less efficiently to the NAD^+^ pool than vitamin B3. Sixty times as much Trp is needed to generate NAD^+^ than is generated from vitamin B3 (as has been reported for niacin-equivalents) [8]. Therefore, it is not likely that Trp is a suitable source for enriching the NAD^+^ pool, also because an overload of Trp can cause toxicity, especially in the central nervous system [9]. The vitamin B3 forms nicotinic acid (NA) and nicotinamide (Nam) are commonly used, but high-dose supplementation can induce negative side effects, including skin flushing by NA and the inhibition of sirtuins by Nam [10]. 

Nicotinamide mononucleotide (NMN) and nicotinamide riboside (NR) have been described as potent NAD^+^ boosters that can activate especially sirtuin 1 (SIRT1) activity [2,3,11,12]. Supplementation with NMN and with NR in mice was shown to induce a beneficial metabolic adaptation which counteracted the metabolic dysfunctions induced by nutrient overload [2,3,4,5]. NMN, an intermediate in NAD^+^ biosynthesis and final precursor to NAD^+^, was shown to be metabolized into either NR or Nam extracellularly in several in vitro studies [13]. Despite the degradation of NMN, administering NMN in vivo has been demonstrated to elevate NAD^+^ levels in multiple tissues [2]. NR appears to be superior to the other NAD^+^ precursors, because of its higher potency to raise NAD^+^ levels in multiple cell lines and tissues [3,13,14]. These promising results subsequently led to the set-up of several human supplementation studies and clinical trials using NR as supplementation product [14,15,16].

White adipose tissue (WAT) is a lipid storage organ, protecting the body against lipotoxicity and providing nutrients in times of need. WAT actively communicates with other organs and secretes peptide hormones—e.g., cytokines and chemokines—that modulate insulin-stimulated glucose metabolisms in other organs such as the liver, muscle and brain, thus playing a crucial role in whole-body glucose homeostasis [17]. WAT dysfunction induced by nutrient overload has been associated with the impaired transcriptional regulation of the NAD^+^/sirtuin pathway and a decreased NAD^+^ level [2,6] and is ameliorated upon vitamin B3 supplementation or calorie restriction in both mice and humans [18]. Therefore, WAT health is likely sensitive to changes in NAD^+^ flux and responsive to exogenous NAD^+^ boosters. In accordance with this, our previous NR dose-response study in mice showed that morphological alterations occurred in WAT upon NR supplementation [19].

Despite the demonstration of beneficial effects of NR supplementation, a limited number of studies have also indicated adverse effects of high-dose NR in rodents on either excise performance [20,21] or metabolic flexibility [19]. In view of the potential human application, the contrasting results warrant a detailed analysis of the metabolic consequences of high-dose NR supplementation on a physiological as well as a molecular level. Therefore, we here investigate the effects of a high dose of NR, in the context of a mildly obesogenic diet, on whole-body metabolic homeostasis, as well as on its molecular and physiological effects in WAT. Since we did not observe beneficial effects of an NR dose of 900 mg per kg of diet (900NR in [19]), which is at the low end of the dose range that displayed NAD^+^ boosting or beneficial effects on metabolic function in other studies [3,4,5,13,14], we here imposed a dose of 9000 mg NR per kg diet (high NR), which is one order of magnitude higher and just at the high end of this dose range. Moreover, blood and tissue samples were collected in a postprandial state for a better correspondence between metabolic effects and exposure of dietary NR.

## 2. Material and Methods

### 2.1. Animals and Diets

The animal experiment was approved by the Animal Welfare Committee of Wageningen University, Wageningen, The Netherlands (DEC2016033.b). This experiment was designed as an independent experiment but performed as part of a larger experiment to reduce the number of (control) animals. Eight or nine-week-old C57BL/6JHsdRcc male mice (Envigo, Horst, The Netherlands) were individually housed (12 h light–dark cycle, 23 ± 1°C, 55 ± 15% humidity), with ad libitum access to feed and water throughout the whole experiment, unless indicated otherwise. A semi-synthetic diet (Research Diet Services, Wijkbij Duurstede, The Netherlands), containing 40% energy from fat was used to resemble the average human consumption of fat in the Netherlands [22]. The diet contained 0.115% L-tryptophan and either 30 or 9000 mg NR as an exclusive source of vitamin B3 (referred to as control NR and high NR, respectively), and the dietary composition is shown in Appendix A. Mice were acclimatized on the control NR diet for two weeks. Next, mice were stratified on body weight into two experimental groups (*n* = 12/group) and received the control NR or high NR diet for 18 weeks. Body weight and feed intake as well as lean and fat mass (by NMR, EchoMRI, Houston, USA) were measured weekly. 

### 2.2. Indirect Calorimetry and Metabolic Flexibility 

Indirect calorimetry was performed in week 14, using a Pheno Master System (TSE Systems, Bad Homburg, Germany) as described [23]. Briefly, mice were individually housed with a steady normal air flow, 12 h light–dark cycle (7:00 h, lights on). Oxygen consumption, carbon dioxide production, activity (infrared beam breaks), and food and drink intake were automatically recorded. After 20 h of adaptation, ad libitum fed mice were monitored for 24 h starting from 7:00 h. Next, the mice were exposed to a fasting–refeeding challenge. For this, the mice were provided with 1.5 g of fresh experimental diet at 16:00 h. The mice fully consumed this, after which they changed to a fully fasted state. The next day, at 16:00 h, they were provided with 1.8 g of fresh experimental diet (refeeding), which was fully consumed. The respiratory exchange ratio (RER) and energy expenditure were calculated by TSE software (TSE systems). Metabolic flexibility was assessed as described [19], with the following modifications: the incremental areas under the RER curve (iAUC) between 22:00 h to 7:00 h during fasting and between 16:00 h to 23:00 h during refeeding were individually analyzed using GraphPadPrism v5.04 (San Diego, CA, USA). The iAUC of RER represents metabolic flexibility. The mean RER of the fasted period was calculated to indicate fatty acid oxidation. 

### 2.3. Oral Glucose Tolerance Test

An oral glucose tolerance test (OGTT) was conducted in week 17. Briefly, feed was removed in the morning and mice were weighed. After exactly 6 h of fasting, a tail cut was conducted, and blood glucose was measured using a Freestyle blood glucose meter (Abbott Diabetes Care, Hoofddorp, The Netherlands) as t = 0 timepoint. Then, 2 g of glucose per kg body weight was administrated via oral gavage. Blood glucose was measured at the timepoints of 15, 30, 60, 90 and 120 min after oral gavage. At t = 0, 15 and 30, 20 µL of blood was collected using Microvette CB 300 tubes (Sarstedt, Etten-Leur, The Netherlands). Plasma was obtained after centrifuging at 2000 g, at 4 °C for 20 min, for insulin measurement.

### 2.4. Sample Collection at Sacrifice 

At the end of the study, mice were provided with a limited amount of diet (0.8 g of fresh experimental diet) at the start of the dark phase so they were fasted at the end of the dark phase; they were subsequently refed with 1.8 g of diet at the start of the light phase for 4 h and then sacrificed by decapitation. The purpose of this refeeding regime was to achieve the maximal discrepancy of RER between two groups, which was seen in the metabolic flexibility assessment. Any remaining feed before dissection was recorded to calculate feed intake. Blood was collected to obtain serum and to measure immediate blood glucose concentrations using a Freestyle blood glucose meter (Abbott Diabetes Care). For serum collection, blood was centrifuged at 3000 g, at 4 °C for 10 min, and was immediately stored at −80 °C. Right epididymal white adipose tissue (eWAT) was rapidly dissected, snap-frozen in liquid nitrogen and stored at −80 °C. Left eWAT was weighed, divided in half and fixed for 24 h at 4 °C in PBS with 4.0% paraformaldehyde (pH = 7.40), as described [19]. Fresh liver was dissected, weighed, snap frozen in liquid nitrogen and stored at −80 °C.

### 2.5. Plasma or Serum Parameters

Plasma insulin was measured using a mouse insulin ELISA kit according to the manufacturer’s instructions (Crystal Chem, Downers Grove, USA). Serum total cholesterol (TC), high-density lipoprotein (HDL) cholesterol, triglycerides (TG) and non-esterified fatty acids (NEFA) were measured using liquicolor enzymatic colorimetric tests (Instruchemie, Delfzijl, The Netherlands), as previously described [19]. Serum low-density lipoprotein (LDL) cholesterol concentrations were calculated using the modified Friedewald formula (LDL cholesterol = total cholesterol - HDL cholesterol – triglycerides × 0.16, for rodents). Serum leptin was measured with a Bio-Plex Pro Mouse Diabetes Assay using a Bio-Plex 200 system (Bio-Rad, Veenendaal, The Netherlands), as published [19]. Serum adiponectin was diluted in 1:10,000 assay buffer and then measured using a mouse adiponectin ELISA kit according to the manufacturer’s instructions (Crystal Chem, Downers Grove, USA). To assess insulin resistance, HOMA-IR was calculated as (fasting glucose (mmol/L)× fasting insulin (mU/L)/14.1) [24].

### 2.6. Liver TG and NAD^+^ Metabolites

Liver TG content was measured using a Triglycerides Liquicolor Kit (Human, Wiesbaden, Germany). In brief, liver tissue was homogenized in ice-cold extraction buffer using an automated pellet mixer (VWR, Boxmeer, The Netherlands), equalizing to 20 mg tissue per ml extraction buffer, followed by sonication 18× (amplitude 40, duty cycle 40, output control 1) (Branson Ultrasonics, Danbury, CT, USA). After a brief vortex, tissue extraction was employed for TG determination in triplicate. Levels of nicotinic acid adenine dinucleotide (NAAD) and N-methyl-4-pyridone-5-carboxamide (Me4py) in the liver were quantified as described [14,25]. 

### 2.7. Gene Expression

RNA was isolated from eWAT using Trizol combined with an RNeasy Mini kit (Qiagen, Venlo, The Netherlands), as described [19]. RNA purity and integrity were verified using Nanodrop (NanoDrop, Wilmington, USA) and TapeStation (Agilent, Santa Clara, CA, USA), respectively. cDNA synthesis, followed by regular qRT-PCR was performed as described [19]. Low-expressed genes were pre-amplified for 12 cycles before qRT-PCR using SsoAdvancedPreAmpSupermix (Bio-Rad). References genes were selected based on stable expression. The expression of each gene was normalized by the reference genes using CFX Manager software (Bio-Rad). Primer sequences and PCR annealing temperatures are shown in Appendix A.

### 2.8. Antibodies

Anti-GLUT4, anti-phospho-AKT (Thr308), and anti-ACTB antibodies were purchased from Abcam (Cambridge, MA, USA); anti-phospho-AKT (Ser473) and anti-AKT antibodies were from Cell signaling (Beverly, MA, USA); anti-MAC2 antibody was from Cedarlane (Ontario, Canada); and rabbit IgG antibody was from Vector Laboratories (Burlingame, CA, USA). Goat anti-rabbit Alexa Fluor 488 secondary antibody was from Life technologies (Carlsbad, CA, USA); goat anti-mouse for ACTB—or otherwise donkey anti-rabbit secondary antibodies—were from LI-COR (Lincoln, NE, USA). 

### 2.9. Western Blot 

eWAT protein extraction and immunoblotting were performed as published [23], with the following modifications. Frozen tissue was homogenized using an automated pellet mixer (VWR) in ice-cold lysis buffer containing complete protease inhibitor cocktail (Roche, Mannheim, Germany), phosphatase inhibitor-Mix I (Serva, Heidelberg, Germany), 2 μM trichostatin A, and 10 mM Nam (Sigma-Aldrich), followed by sonication 18× (amplitude 40, duty cycle 40, output control 1). The lysate was centrifuged at 18,620 g, at 4 °C for 15 min; then, the supernatant was retrieved and centrifuged again with the same procedure. The obtained clear lysate was employed to measure protein concentration using DC protein assay (Bio-Rad), mixed with LDS loading buffer and dithiothreitol, heated at 70 °C for 10 min, briefly centrifuged and run on a 4–12% Bis-Tris gel (Invitrogen, Carlsbad, CA, USA) at 110 V for 40 min, then 150 V for 50 min. The protein was transferred to an Immobilon PVDF membrane (Merck Millipore, Amsterdam, the Netherlands) at 300 mA for 1 h, after which the membrane was blocked with 5% BSA in TBS containing 0.1% tween (TBSt) at RT for 1 h and then incubated with primary antibody at 4°C overnight. After 6 times washes using TBSt, the membrane was incubated with secondary antibody at RT for 1 h. The membrane was washed using TBS and scanned on an Odyssey scanner (LI-COR). Bands were analyzed using Odyssey software V3.0 (LI-COR).

### 2.10. Histology and Immunohistochemistry 

The adipocyte cell size determination, number calculation as well as macrophage staining and counting were performed as described previously [19]. Briefly, the tissue was fixed, washed in PBS, embedded in paraffin and sectioned at 5 µm using an automated microtome (Microm GmbH, Heidelberg, Germany). Sections after 20 sequential cuts were used to ensure no repetitive adipocytes were present. Tissue sections were deparaffinized, rehydrated and then stained with Mayer’s hematoxylin for 75 s (Klinipath BV, Duiven, The Netherlands) and with 0.1% eosin for 15 s (Brunswig, Southborough, MA, USA). Subsequently, sections were dehydrated, mounted and then dried overnight at 37 °C. Representative pictures were taken using a Leica DM6B microscope (Leica Microsystems, Wetzlar, Germany). The adipocyte size was measured using CellProfiler software v2.1.1 (The Carpenter Lab, Massachusetts, USA) and expressed in surface area (μm^2^) per adipocyte. The frequency distribution of adipocyte size was calculated as described [19] with some modifications. Briefly, the adipocyte surface area was distributed in 100 μm^2^ clusters in Excel and subsequently clustered as defined fractions of small (100–1500 μm^2^), medium (>1500–6000 μm^2^), and large (>6000 μm^2^). Values less than 100 μm^2^ were excluded. Macrophages were stained using anti-MAC2 antibody from Cedarlane (Ontario, Canada). Single macrophages and crown-like structures (CLSs) from 1000 adipocytes per animal were counted and expressed as macrophages or CLSs per gram eWAT weight.

Fluorescence microscopy of glucose transporter type 4 (GLUT4 or SLC2A4) was conducted as follows. Sections were deparaffinized, rehydrated and the heat-induced epitope was retrieved in 10 mM sodium citrate buffer (pH = 6) for 10 min. Sections were subsequently blocked using 5% normal goat serum in PBS-BSAc (S-1000, Vector Laboratories, CA, USA) at RT for 1 h, and then incubated with GLUT4 antibody (1:500) or rabbit IgG antibody at 4 °C overnight. After six washings using PBS, sections were incubated with Alexa Fluor 488 antibody (1:200) at RT for 1 h, followed by DAPI staining (Sigma-Aldrich, Steinheim, Germany) for 5 min. Sections were then mounted and kept at 4 °C in the dark. Representative pictures (*n* = 15 per animal plus a negative control) were taken at 20× magnification using a Leica DM6B microscope (Leica microsystems, Wetzlar, Germany) and at 100× magnification using a Zeiss Axioscope 2 microscope (Zeiss, Munich, Germany), with 350 nm (DAPI) and 475 nm (FITC) channels. Staining and microscopy were performed at the same time for control NR and high NR, with identical settings.

Quantification of GLUT4 fluorescence intensity at 20× magnification was performed using ImageJ software as follows. The GLUT4 fluorescence of the whole area of each image (*n* = 15 images per animal) was quantified, corrected by the background of each image and the negative control. The nuclei of each image were counted based on binary images of DAPI. GLUT4 intensity was expressed as the corrected integrated density by the number of the nuclei.

### 2.11. Statistics

Statistical analysis was performed using GraphPad Prism v5.04 and indicated in the figure legends. Data were verified for normality using the D’Agostino and Pearson omnibus normality test and log transformed if needed. *P*-values <0.05 were considered to be statistically significant. Regression analysis for the relation between iAUC of RER and body weight, lean mass, or between mean RER and activity was performed using proc MIXED in SAS 9.4 (Cary, NC, USA) with the treatment included in the model. Interactions with treatment were never significant and were removed from the models.

## 3. Results

### 3.1. High Dose NR Reduces Metabolic Flexibility in HF Diet-Fed Male C57BI/6JRccHsd Mice

Since dietary NR was shown in multiple animal studies to improve metabolic health [3,5,11,12], but also was shown to either have no effect [26] or have detrimental effects depending on the dose and animal model used [19,20,21], we analyzed the long-term (18 weeks) metabolic health effects of dietary high doses of NR in HF diet-fed male C57BI/6JRccHsd mice. We compared high NR to control NR, unless indicated otherwise. We monitored body weight, feed intake, fat mass, and lean mass weekly. We did not observe significant differences in body weight and feed intake over the whole 18-week period (Figure 1a,d). Whole body fat mass tended to be higher in the first 6 weeks during NR treatment (*p* = 0.146), but was normalized after 18 weeks (Figure 1b), while lean mass did not differ (Figure 1c). We sacrificed the animals after a refeeding to be able to analyze plasma and tissue metabolic parameters in a postprandial state. No difference was seen in refeeding feed intake (Appendix A). We observed a decrease in eWAT weight in the high NR group (Figure 1e), concomitant with an increase in both liver weight (Figure 1f) and liver TG (Figure 1g). Levels of NAAD and Me4py—the biomarkers for NR exposure effects—were increased in the liver by the high NR treatment (Figure 1h,i). Furthermore, we also found a significant elevation of serum TC and a tendency to elevated LDL cholesterol (*p* = 0.053) in the high NR-treated mice, whereas other circulating markers of lipid metabolism (TG, NEFA, HDL cholesterol, adiponectin and leptin) and blood glucose were not different (Appendix A). 

To further study the metabolic impact of high doses of dietary NR, we analyzed whole-body metabolic flexibility using indirect calorimetry. Previously, we showed by using a fasting–refeeding challenge that a dose of 900 mg NR per kg diet compromised metabolic flexibility [19]. Performing the same fasting–refeeding challenge now on a 9000 mg NR per kg of diet again demonstrated a decrease in metabolic flexibility by high-dose dietary supplementation with NR (Figure 1j). High NR significantly reduced the change of RER during the transition from food withdrawal to a physiologically fasted state (when RER reaches 0.7) (Figure 1k) as well as during refeeding (Figure 1l). High NR did not alter RER in the fasted period, where the RER for both groups was near 0.7 (Figure 1m). No differences were observed in energy expenditure or physical activity (Appendix A). No significant correlation nor interaction were found between treatments for the iAUC of RER and body weight or lean mass, or between mean the RER and activity during the refeeding period (Appendix A).

### 3.2. High NR-Fed Mice Have Lower Glucose Tolerance 

The observed decrease in metabolic flexibility in mice fed high NR was most prominent in the first few hours during the refeeding transition from fat oxidation to carbohydrate oxidation (Figure 1j), indicating that postprandial carbohydrate metabolism might be hampered by high NR. To examine this in more detail, we analyzed how high NR-treated mice responded to an oral glucose bolus via an OGTT. We observed a decreased glucose clearance rate (Figure 2a,b), which was accompanied by an impaired insulin response (Figure 2c,d) in the high NR-treated mice during OGTT. Compared to the control group, the fasting plasma insulin level was elevated almost 2.5-fold in mice fed high NR (Figure 2e), leading to a much higher HOMA-IR index (Figure 2f). Combined, these results show that treating HF diet-fed mice with a high dose of NR lowers glucose tolerance and aggravates systemic insulin resistance.

### 3.3. High NR Feeding Results in a Lowered Peroxisome Proliferator-Activated Receptor γ (PPARγ) Expression Signature in WAT

Since high NR-fed mice were insulin resistant and also showed a decreased weight of eWAT depots, we aimed to analyze whether WAT from the high NR-treated mice was metabolically dysfunctional. PPARγ is widely known as a master regulator of WAT functions, including insulin sensitivity, adipogenesis and inflammation [27]. Therefore, we analyzed the gene expression of PPARγ and its target genes in the eWAT. Pparg gene expression was significantly reduced upon high NR feeding (Figure 3a). Concomitantly, PPARγ-target genes in insulin signaling (Adipsin, Grb14, Glut4) and glyceroneogenesis (Pck1, Scd1) were regulated. All PPARγ-positively regulated genes were downregulated (Adipsin, Scd1, and Pck1) or tended to be downregulated (Glut4), whereas the negative regulator of the insulin receptor, Grb14, which is physiologically downregulated upon PPARγ activation, was upregulated (Figure 3b). This is in line with a downregulation of a the PPARγ expression signature in WAT upon high NR feeding. 

The regulation of insulin resistance in WAT by PPARγ is accompanied by alterations of several components of insulin signaling and is often co-regulated. Although we did not find differences in the gene expression of Insr, Irs1 or Irs2 (Appendix A), the phosphorylation of AKT at both the Thr308 and Ser473 sites was decreased upon high NR feeding in the eWAT (Figure 3c–e), whereas total AKT levels were not different. A lower AKT phosphorylation results in decreased downstream GLUT4 translocation [28].

GLUT4 expression in WAT is crucial for whole-body glucose disposal and has been shown to be increased upon PPARγ activation or decreased due to PPARγ repression [29,30]. In rodent and human adipocytes, regulation of Glut4 mRNA level influences GLUT4 protein levels [31]. Since *Glut4* gene expression was decreased upon high-dose NR, but did not reach statistical significance (*p* = 0.051), we analyzed the expression of GLUT4 protein levels. GLUT4 protein expression was observed in the adipocytes (Figure 4a), and the intensity of GLUT4 staining was significantly reduced in the eWAT from mice treated with high NR (Figure 4b,c). Combined, the lowered PPARγ expression signature, decreased AKT phosphorylation and lower GLUT4 expression in eWAT demonstrate on the molecular level that the eWAT of high-dose NR treated HF-diet mice is less insulin responsive.

### 3.4. High-Dose NR Leads to More Severe WAT Inflammation 

Despite reduced eWAT weight in the high NR group (Figure 1e), we did not observe any differences in the mean adipocyte size (Figure 5a,b) or frequency distribution of adipocyte sizes (Appendix A). Insulin resistance is often associated with higher WAT inflammation, characterized by a higher expression of pro-inflammatory markers and macrophage infiltration or aggregation (presented as crown like structures (CLSs)). First, using specific staining for macrophages in WAT sections, we were able to analyze macrophage infiltration and aggravation (Figure 5c). An increased number of single macrophages was observed in sections of WAT from high NR-fed mice (Figure 5c,d). Furthermore, the number of CLSs was also increased upon high NR feeding (Figure 5c,e). Second, using qRT-PCR, we found a marked upregulation of pro-inflammatory genes, including the M1 macrophage markers (*C*d11c and *C*d11d) as well as those in an acute-phase response (C3, S1008a, Saa1 and Saa3) and in inflammasome activation (Casp1) in the WAT of high NR-fed mice (Figure 5f). Collectively, these results demonstrate that high NR feeding induced pro-inflammatory gene expression and macrophage infiltration in WAT.

## 4. Discussion

The metabolic health of C57BI/6JRccHsd mice fed a high dose NR diet (9000 mg NR/kg diet) was deteriorated in our study, since metabolic flexibility was reduced, the glucose clearance rate was lower, and insulin resistance on the whole-body level was aggravated. eWAT depot weights were decreased by high NR feeding, whereas liver weight and liver TG were increased. We found a lowered PPARγ expression signature, a downregulation of AKT/GLUT4 signaling, and aggravated pro-inflammatory responses in the eWAT of mice on thehHigh NR diet. Altogether, these alterations in mice treated with high-dose NR suggest the onset of WAT dysfunction, which may be in part associated with the deterioration of metabolic health. 

Metabolic flexibility on the whole-body level is defined as the ability to readily switch between carbohydrate oxidation and fatty acid oxidation in response to a physiological or nutritional intervention [32]. We observed a decreased metabolic flexibility with a high dose of NR treatment, in agreement with our previous finding that mice fed a mildly obesogenic diet containing a moderately high dose of NR (900 mg NR per kg of diet) were less metabolically flexible compared to those with NR at the recommended vitamin B3 level [19]. In contrast, Canto et al. found that metabolic flexibility, assessed by the subtraction of mean RER of dark phase from those from light phase, was increased by supplementation with 400 mg NR per kg BW per day in mice fed a HF diet [3]. This increase in metabolic flexibility, however, was not seen in mice fed a high-fat–high-sucrose (HFHS) diet [4], nor in mice fed a low-fat diet [11], even though a similar high dose of NR was used. This suggests a modifying effect of the obesogenic diet on the improvement of metabolic flexibility at high NR doses. A change in metabolic flexibility implicates altered metabolic homeostasis [33]. Our data imply that whole-body lipid homeostasis may remain unchanged in response to high-dose NR, since the whole-body fat mass, the fasted RER, as well as the postprandial circulating TG and NEFA levels were similar between groups (Figure 1b,m and Appendix A). On the other hand, our observations during an OGTT suggest that glucose homeostasis was impaired by high-dose NR (Figure 2). Altogether, these data suggest that a worsening of glucose homeostasis is likely a main feature of high-dose NR-induced deterioration of metabolic health in our study.

WAT plays a critical role in whole-body glucose disposal. A defect of insulin signaling in WAT can have a significant impact on glucose homeostasis [34]. The observed downregulation of AKT/GLUT4 signaling in eWAT induced by high NR treatment (Figure 3 and Figure 4) implies that the WAT of the high dose NR-treated mice is likely less capable of maintaining glucose homeostasis. The phosphorylation of AKT in adipocytes is regulated by distinct players in insulin signaling pathways; for example, phosphoinositide-dependent protein kinase 1 (PDPK1) induces Thr308 phosphorylation [28], and mechanistic target of rapamycin complex 2 (mTORC2) mediates Ser473 phosphorylation [28]. Possibly, the concurrent regulation of Thr308 and Ser473 phosphorylation of AKT in the response of WAT to high-dose NR implicates PDPK1 and mTORC2 in the regulation of insulin signaling. GLUT4 is thought to be the major glucose transporter in WAT, serving as a downstream effector of insulin signaling cascade mediating glucose uptake [34]. AKT activation ultimately leads to GLUT4 translocation, and the maximal activity of AKT for this event requires phosphorylation of both Thr308 and Ser473. It is conceivable that the downregulation of Thr308 and Ser473 phosphorylation at both sites by high NR treatment may lead to depressed GLUT4 translocation. Furthermore, high NR treatment resulted in an almost two-fold downregulation of GLUT4 expression on the transcriptional and protein levels in eWAT, indicating an impairment of WAT glucose handling. In support of this notion, adipocyte-specific GLUT4 knockout mice also displayed glucose intolerance and systemic insulin resistance [35]. 

A lowered PPARγ expression signature in eWAT induced by High NR treatment may also contribute to a worsening of glucose homeostasis and insulin resistance. This signature was composed of the downregulation of Pparg, adipsin, Glut4, Scd1 and Pck1 and the upregulation of Grb14 (Figure 3a,b). Adipsin gene expression is tightly negatively correlated with PPARγ phosphorylation at Ser237 in white adipocytes [36]. Downregulation of the adipsin mRNA level is therefore in full agreement with the increased PPARγ phosphorylation in WAT of the high NR-treated mice. Adipsin (complement factor D, CFD) is a crucial adipokine that can improve insulin secretion and glucose homeostasis [37]. Adipsin gene expression in WAT and circulating adipsin levels were found to be dramatically decreased in ob/ob and db/db mice, probably associated with the diabetic phenotypes in those models [38]. GRB14 binds to the activated insulin receptor and inhibits its catalytic activity, thus blocking insulin signal transduction [39]. *Grb14* is upregulated in the WAT of insulin-resistant mice and type 2 diabetic patients [40]. *Pck1* encodes the gluconeogenic enzyme PCK1, but in WAT, this enzyme is more glyceroneogenic [41]. SCD1 catalytically de-saturates fatty acyl–CoA substrates and also plays a role in glyceroneogenesis [42]. Targeted deletion of Scd1 in adipocytes impaired glyceroneogenesis in WAT, concomitant with the reduced other glyceroneogenesis markers, including the downregulation of Pck1 [42]. Dysregulation of glyceroneogenesis in WAT induced by Pck1 mutation resulted in elevated TG levels in liver [43], impaired WAT glucose uptake and global glucose intolerance [43], a similar metabolic phenotype as seen in high dose NR-treated mice in our study. 

A high dose of NR aggravated pro-inflammatory responses in eWAT both on the morphological level and the molecular level (Figure 5c–f), implicating WAT pro-inflammation in whole-body insulin resistance. The typical M1 macrophage markers, *Cd11c* and *Cd11d*, were transcriptionally increased, indicating that more M1 macrophages were present. WAT M1 macrophages generate pro-inflammatory cytokines, such as tumor necrosis factor α (TNFA-α) and interleukin 6 (IL6), thus contributing to the induction of insulin resistance. *C3,* Saa1 and Saa3, which is regulated by *S100a8* [44], encode secretory proteins that can excrete from adipocytes into the local environment, leading to the enhanced inflammation in WAT and insulin resistance [45,46,47]. Interestingly, these four genes have been reported to be downregulated in response to PPARγ-agonist treatment by us [48] and others [45,46,49], in agreement with the amelioration of inflammation. The upregulation of *C3, Saa1*, Saa3, and S100a8 aligns with the observed lowered PPARγ expression signature. Casp1 expresses Caspase 1, which governs the production of the pro-inflammatory cytokine IL1β, playing a role in the development of insulin resistance [50].

WAT serves as a dynamic storage organ for the excessive lipid load that can lead to WAT expansion, characterized by the adipocytes increase in both number (hyperplasia) and size (hypertrophy). The characteristics of adipocyte size are comparable (Figure 5a,b and Appendix A), suggesting that high-dose NR did not affect WAT expansion towards hypertrophy. However, considering that tissue weight was lower (Figure 1e), the total number of adipocytes of the whole tissue might be also lower, indicating that obesogenic diet-induced WAT expansion towards hyperplasia was, to some extent, inhibited by a high dose of NR. Furthermore, the increased number of CLSs in WAT of the high-dose NR-treated mice suggests enhanced adipocytes cell death. Taken together, it is plausible that fewer functional adipocytes were available for lipid or cholesterol storage, which could potentially lead to the increased ectopic deposition of lipids in other metabolic organs [51]. Thus, we observed that serum TC (Appendix A) and liver TG levels (Figure 1g) were increased by a high dose of NR.

Three studies have shown that supplementation with a high dose of NR improves glucose homeostasis in C57BL/6J mice fed either a HF diet or a HFHS diet [3,4,5]. In our C57BL/6JRccHsd mouse strain, we observed impaired glucose homeostasis upon treatment with a similar high dose of NR and a mildly obesogenic diet. One could argue that the inconsistency between studies could be due to a hormetic effect of NR; i.e., a lower dose than high NR (9000 mg NR per kg diet) may result in beneficial effects on metabolic health. We consider this to be unlikely, because we previously observed a decreased metabolic flexibility in mice on 900 mg NR per kg of diet—a dose one order of magnitude lower than the dose used in the present study. Furthermore, as was seen for high NR, 900 mg NR per kg diet increased liver weight and tended to elevate hepatic TG levels, and in eWAT significantly decreased gene expression of adipsin and did not improve pro-inflammatory gene expression (Appendix A). In addition to a possible modifying effect of the obesogenic diet, the genetic differences in mice models may also play a role. The C57BL/6JRccHsd mouse strain in our study contains a functional nicotinamide nucleotide transhydrogenase (Nnt) gene, whereas the C57BL/6J model that was used in the studies showing beneficial effects of high-dose NR supplementation harbors a spontaneous mutation in *Nnt,* which has been described to be responsible for an impaired glucose tolerance and decreased insulin secretion in this model [52,53]. More importantly, NNT is a mitochondrial inner membrane-located protein that catalyzes the reversible transfer of hydrogen between NAD^+^ and NADP^+^, thus contributing to NAD^+^ homeostasis. It has been shown that the loss of *Nnt* decreases the NAD^+^/NADH ratio, whereas NNT overexpression increased this ratio [54]. This implies that the functional effects of NR supplementation may depend on the intactness of redox homeostasis and/or NAD^+^ homeostasis. Other models with a functional Nnt gene did not show a beneficial effect of high-dose NR supplementation. For example, supplementation with 300 mg NR/kg BW/day impaired redox homeostasis in the skeletal muscle of young rats, in a model containing functional *Nnt* gene, leading to decreased exercise performance [20,21]. It is therefore of interest to examine the metabolic effects of high-dose NR supplementation in genetically identical mice with and without *Nnt*, as well as in models of impaired redox homeostasis. It is also important to investigate the insulin signaling in the skeletal muscle in response to high-dose NR, considering the fact that skeletal muscle may account for the majority of insulin-stimulated glucose uptake.

To conclude, a high dose of NR induced the deterioration of metabolic health, which was characterized by impaired glucose homeostasis and insulin resistance in C57Bl/6JRccHsd mice. The underlying mechanisms may be associated with molecular and morphological changes related to WAT dysfunction, including decreased AKT/GLUT4 signaling, a lowered PPARγ expression signature, and aggravated pro-inflammation. Nnt may potentially explain the metabolic effects induced by exposure to a high dose of NR. Its absence may possibly explain the beneficial effects reported in various other studies. Considering the fact that NNT is generally expressed and functional in humans [55] as well as the increasing concerns regarding the validity of the C57BL/6J model for human diabetes research [52,53], further investigation using the proper models is warranted to fully understand the biological functions of NR before more human trials are performed.

## Figures and Tables

**Figure 1 nutrients-11-02439-f001:**
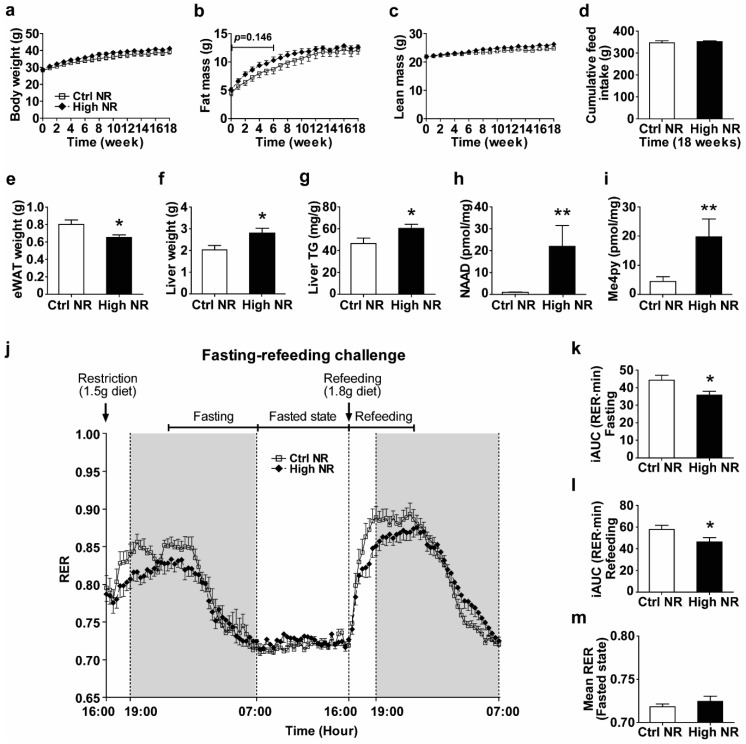
A high dose of nicotinamide riboside (NR) reduces metabolic flexibility and epididymal white adipose tissue (eWAT) weight. Mice were on obesogenic diets containing either 30 mg vitamin B3 (nicotinamide riboside, NR) per kg diet (control NR) or 9000 mg per kg diet (high NR) for 18 weeks. Body weight (**a**), fat mass (**b**), lean mass (**c**), cumulative feed intake (**d**). Fresh tissue weight of eWAT (**e**) and liver (**f**). Liver triglyceride (TG) content was determined and expressed per gram of tissue (**g**). Biomarkers for NR exposure effects are represented by the metabolites nicotinic acid adenine dinucleotide (NAAD) (**h**) and Me4py (**i**) in the liver. Levels are in pmol per mg frozen pulverized tissue. Fasting–refeeding challenge (**j**): restriction with 1.5 g of diet at 16:00 h, fasting (from 22:00 h to 7:00 h), fasted (from 7:00 h to 16:00 h), refeeding with 1.8 g of diet at next 16:00 h, refeeding (from 16:00 h to 23:00 h) periods. Shaded areas indicate the dark, active periods. The incremental areas under the curve (iAUC) of the respiratory exchange ratio (RER) during fasting (**k**) and during refeeding (**l**) were analyzed to represent metabolic flexibility. The mean RER of the fasted state (**m**). Control NR: open square with solid line or white bar; high NR: closed diamond with black dashed line or black bar. Data are analyzed using either two-way repeated measures ANOVA followed by Bonferroni post-hoc analysis (**a**–**c**,**j**) or Student’s *t*-test (**d**–**i**,**k**–**m**), and presented as mean ± SEM (*n* = 11–12 mice per treatment). * *p* < 0.05, ** *p* < 0.01.

**Figure 2 nutrients-11-02439-f002:**
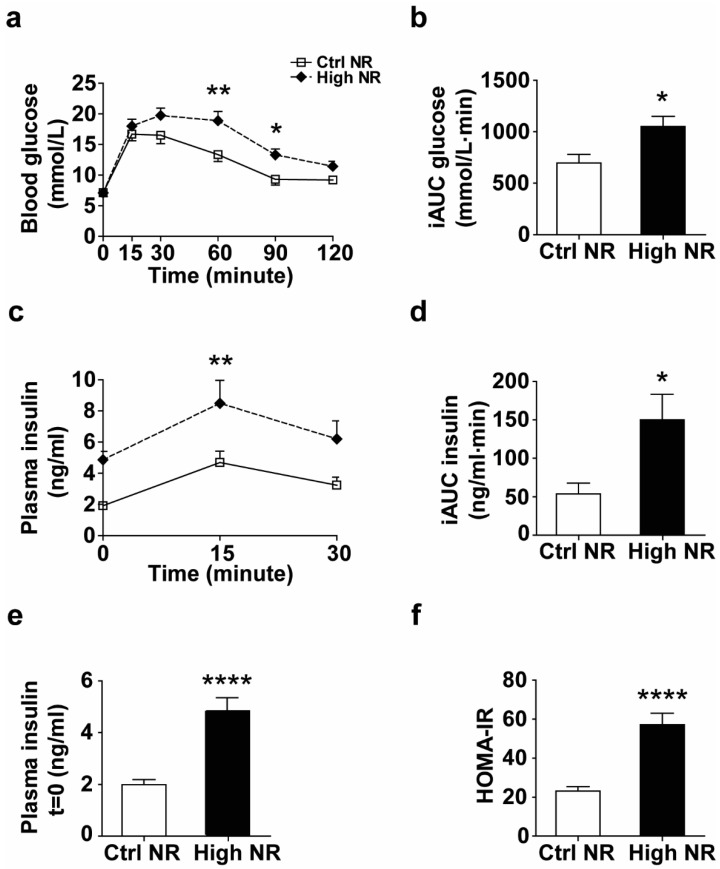
High NR induces lower glucose tolerance and higher circulating insulin levels during an oral glucose tolerance test (OGTT). Blood glucose was measured at indicated timepoints before and after an oral glucose administration (**a**) and iAUC was calculated to assess glucose tolerance (**b**). Plasma insulin was measured at indicated timepoints during OGTT (**c**) and iAUC (Baseline = 1.994) was analyzed (**d**). Plasma insulin at *t* = 0 represents fasted circulating insulin (**e**) and the corresponding HOMA-IR was calculated to assess insulin resistance (**f**). Control NR: open square with solid line or white bar; high NR: closed diamond with black dashed line or black bar. NR in mg/kg diet. Data are analyzed using either two-way repeated measures ANOVA followed by Bonferroni post-hoc analysis (**a**,**c**) or Student’s *t*-test (**b**,**d**–**f**), and presented as mean ± SEM (*n* = 11–12 mice per treatment). * *p* < 0.05, ** *p* < 0.01, **** *p* < 0.001.

**Figure 3 nutrients-11-02439-f003:**
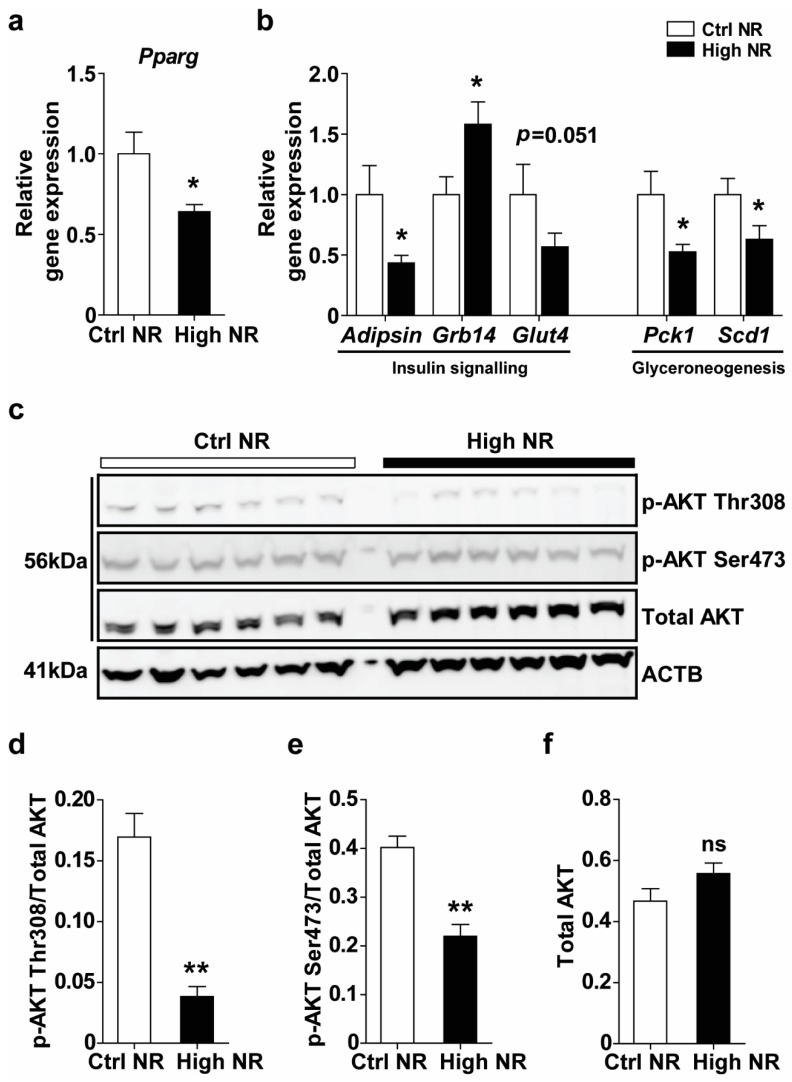
High NR feeding results in a lowered proliferator-activated receptor γ (PPARγ) expression signature. Relative gene expression of PPARγ (**a**) and its targeted genes (**b**) (normalized to the reference genes, *n* = 10–12 per treatment). Phosphorylation and total AKT in eWAT were measured by immunoblotting, using ACTB as a loading control (**c**). Densitometry analysis on the ratio of p-AKT Thr308/AKT (**d**), p-AKT Ser473/AKT (**e**) and total AKT/ACTB (**f**) (*n* = 6 mice per treatment). Control NR: white bar; high NR: black bar. Data are analyzed using Student’s *t*-test (**a**,**b**) or Mann–Whitney test (**d**–**f**) and presented as mean ± SEM. * *p* < 0.05, ** *p* < 0.01.

**Figure 4 nutrients-11-02439-f004:**
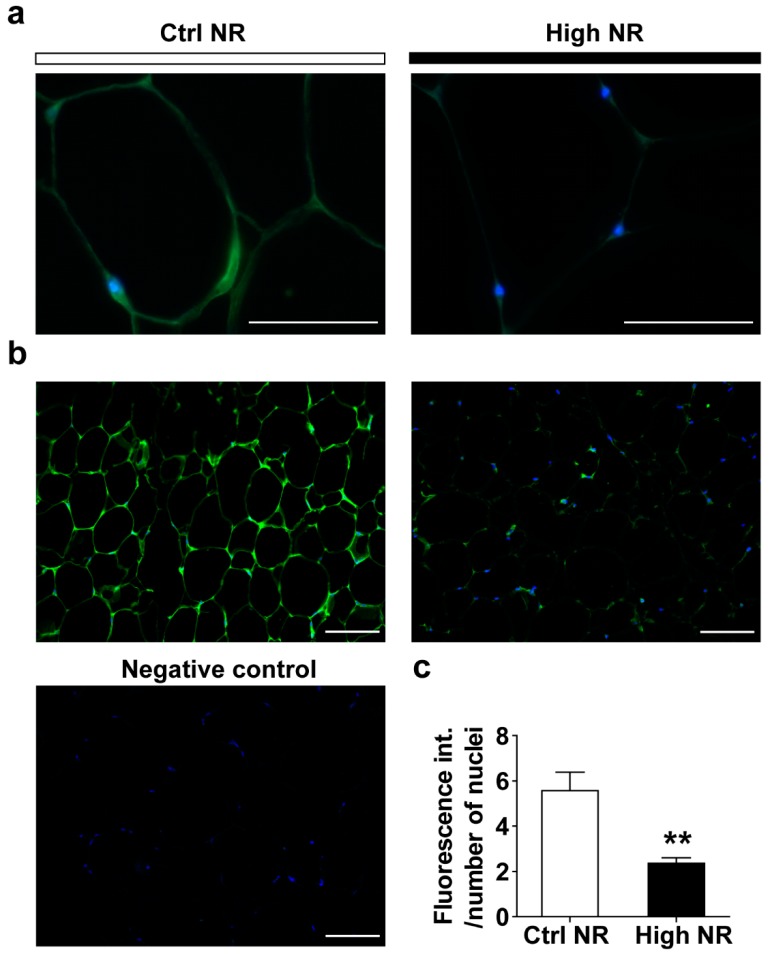
High NR results in lower glucose transporter type 4 (GLUT4) expression in eWAT. Representative pictures of GLUT4 expression at 100× magnification (**a**): Control NR, left; high NR, right. Representative pictures of GLUT4 expression at 20× magnification (**b**): Control NR, left; high NR, right. Negative control from control NR animal, left bottom. Green staining represents GLUT4, blue dots are nuclei. Quantification of GLUT4 fluorescence intensity, normalized by the number of the nuclei (C, *n* = 15 pictures at 20× magnification per animal, six animals per treatment). Scale bar represents 50 µm. Data are analyzed using Mann–Whitney test and presented as mean ± SEM. ** *p* < 0.01.

**Figure 5 nutrients-11-02439-f005:**
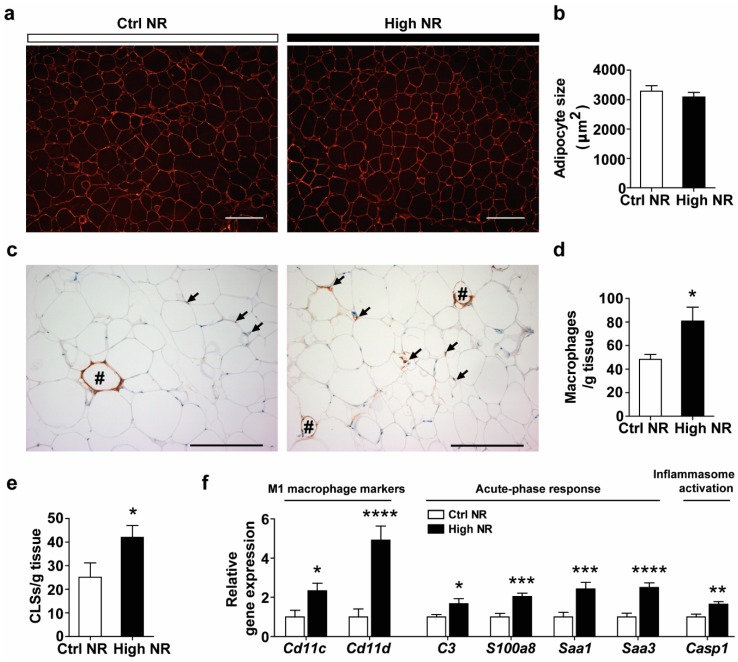
High NR impairs visceral WAT function, characterized as more severe inflammation. Representative images of cell size (**a**) and mean adipocyte size (**b**). Representative images of crown like structures (CLSs) (indicated by hashes) as well as single macrophages (indicated by arrows). Scale bar represents 200 µm. Single macrophages and CLSs were counted (*n* = 8 animals per treatment) and expressed as macrophages or CLSs per gram eWAT weight (**d**,**e**). Relative gene expression (normalized to the reference genes) of pro-inflammatory genes (**f**). Control NR: white bar; high NR: black bar (*n* = 11–12 per treatment). Data are analyzed using Student’s *t*-test and presented as mean ± SEM. * *p* < 0.05, ** *p* < 0.01, *** *p* < 0.005, **** *p* < 0.001.

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
