# Peer review of "High Dose of Dietary Nicotinamide Riboside Induces Glucose Intolerance and White Adipose Tissue Dysfunction in Mice Fed a Mildly Obesogenic Diet"

_nutrients, 2019, doi:10.3390/nu11102439_

Round 1

Reviewer 1 Report

On the surface, it strikes me as a strange question – basically it is “is overdosing on something bad?” They already state that the “low dose” is actually the recommended dose. But this is a recommended dose for humans, not mice. However it is actually relevant in terms of mechanistic NR studies which usually use massive doses in mice. Perhaps forcing that much NR in isn’t a good idea when you are looking at whole body metabolism, because it has negative off target effects.

The high NR diet has 9000 mg/ kg in it and the low one has 30 mg/kg.  The mice ate about 340g food over 18 weeks, so that works out at 2.7 g/day. If we take the average body weight to be about 35g over the full 18 weeks, this works out at about 685 mg/kg/day  for high NR and 2.3 mg/kg/day for low NR.

In short authors saw increased liver weight, reduced metabolic flexibility and white adipose tissue dysfunction with increased inflammation and insulin resistance/impaired glucose tolerance with high dose NR.

Mice were collected after refeeding to look at the postprandial state. This showed decreased RER after fasting and also refeeding thus lowered metabolic flexibility in the high NR group – and they controlled for activity and energy expenditure. They also showed a very significant decrease in glucose tolerance and insulin resistance in high NR fed animals.

Overall body weight and fat mass does not change, but eWAT size decreases. They think that high NR dose causes WAT dysfunction resulting in the pathological deposition of fat in other organs like the liver, backed up with some mechanistic stuff like reduced GLUT4 expression and PPARg reduction.

These data are at odds with the 2012 Canto paper that said NR improves HFD response and glucose tolerance in which a relatively high NR dose was used.

Comments that may help the manuscript:

The calorimetry data could be presented as linear regression/ANCOVA, particularly for RER because they claim there’s a difference there, and it would be useful to see if it is the lean mass or body weight of each mouse is affecting it. We don’t know if mass and/or lean body mass has a different effect on the high NR group compared to the control group, so can’t draw sound conclusions. Similarly, there is a lot of variability in the activity of the control NR group, even though the difference isn’t statistically significant. This may be affecting things in that the most active ones have much higher RER and therefore skew the data – this is why they need to present it as linear regression for each mouse (in this case I would do RER vs activity). Could any differences be because NR gets degraded into NAM in the gut? How stable is NR when mixed into their diet? How often did they change the food? This was not controlled for as far as I can see (although perhaps they don’t care and are only coming from a supplementation point of view). They need to repeat this by feeding NAM/and or ADPR and see if they get the same results. Perhaps too much work though – a potential follow-up. Statements such as “these results show that treating HF-diet fed mice with a high dose of NR lowers glucose tolerance and aggravates systemic insulin resistance” are used. However, the title denotes the diet as “mildly obesogenic”, not high fat. It is only 40% caloric fat which is higher than normal but they only put on 10 grams over 18 weeks which happens as mice get older anyway, and the RER still looks pretty flexible to me. A normal chow diet control would have helped. Why didn’t they look at brown fat at all, with it being a more metabolically active tissue? They cite functional NNT as a possible reason for differences between theirs and the Canto studies. In the discussion they sayWe propose a role for Nnt in the metabolic effects induced by exposure to a high dose of NR”. This is a somewhat spurious claim to make based on an observation which they haven’t substantiated. Liver is bigger, more triglycerides -> also signs of inflammation and fibrosis? -> show images of liver (micro- , macrosteatosis?) AKT phosphorylation is decreased, InsR gene is not regulated. What about InsR and IRS1/2 phosphorylation? Would be interesting to know at which level insulin signaling becomes impaired -> upstream of AKT? Western blot image for phosphoAKT Ser473 not very convincing Cholesterol metabolism seems to be disturbed…effect of NR on cholesterol biosynthesis pathway? NR as potential HMG-CoA reductase inhibitor? (Holdsworth ES, Kaufman DV, Neville E. A fraction derived from brewer's yeast inhibits cholesterol synthesis by rat liver preparations in vitro. Br J Nutr. 1991 Mar;65(2):285-99. ) Casp-1 is not really a marker for apoptosis -> rather activation of inflammasome for apoptosis detection would be interesting to see cleaved caspase-3 or some marker for mitochondrial apoptosis induction (cytochrome c release, AIF, BAX, BAD, other members of Bcl2 family)

Reviewer 2 Report

In the manuscript "High dose of dietary nicotinamide ribose induces glucose intolerance and white adipose tissue dysfunction of mice fed a midly obesogenic diet", the authors demonstrate the role of NR on WAT dysfunction.

I have no major criticisms and the data appears consistent with previously published results, although there are some similar results presented previously by the same authors(Effects of a wide range of dietary nicotinamide riboside (NR) concentrations on metabolic flexibility and white adipose tissue (WAT) of mice fed a mildly obesogenic diet. W Shi, MA. Hegeman, D A.M. van Dartel, J Tang, M Suarez, H Swarts, B Hee, L Arola, J Keijer. Mol Nutr Food Res. 2017 Aug; 61(8): 1600878)

It would be good to better understand the precise mechanism of nicotinamide ribose in inflammation, does the high dose of NR increase proinflammatory proteins due to toxicity?

In Figure 3, molecular weight of pAKT 308 is 55 kDa, pAKT 473 is 60 KDa. Western blot images of p- AKT 473 and total AKT look the same image with different intensity, the membrane has been re-incubated?

Too many references are found in the test (105), the authors should reduce the bibliography.

Line 185 change “antibodieswere”

Reviewer 3 Report

The authors studied the effects of high doses of NR using mice. Previous studies of other grroup have reported only useful health benefits of NR. On the other hand, the authors have shown data that excess NR can be detrimental to health. The authors' reports are valuable information because they suggest important issues to encourage the proper use of NR.

I agree to publish their research report, but I would like them to revise the manuscript before accepting it.

Comment #1

The authors mainly analyze adipose tissue to elucidate the molecular mechanism, but I should mention the effects on liver and muscle tissue.

#1-a) Is there a possibility that an excessive amount of NR is adversely affecting the liver?

#1-b) The authors showed that excessive NR caused glucose metabolism abnormalities, particularly insulin resistance, in the OGTT study. Impaired glucose tolerance should be more dependent on skeletal muscle insulin resistance than adipose tissue. Therefore, authors should either present test results for skeletal muscle status or discuss effects on skeletal muscle.

Round 2

Reviewer 1 Report

I am happy with the quality of the rebuttal addressing all points. 1 minor points, that likely don't require changing manuscript but useful anyway:

In the response they say:

"we would like to point out that the dose of Ctrl NR treatment, namely 30 mg NR per kg diet, is based on the recommended level of vitamin B3 for rodent diets (AIN93, American Institute of Nutrition)."

Well yes but how much of that is NR rather than NA, NAM etc? Not much I'd wager.